# Sexual Orientation Discrimination and Exclusive, Dual, and Polytobacco Use among Sexual Minority Adults in the United States

**DOI:** 10.3390/ijerph19106305

**Published:** 2022-05-23

**Authors:** Delvon T. Mattingly, Andrea R. Titus, Jana L. Hirschtick, Nancy L. Fleischer

**Affiliations:** 1Center for Social Epidemiology and Population Health, Department of Epidemiology, School of Public Health, University of Michigan, Ann Arbor, MI 48109, USA; janahirs@umich.edu (J.L.H.); nancyfl@umich.edu (N.L.F.); 2Department of Population Health, NYU Grossman School of Medicine, New York, NY 10016, USA; andrea.titus@nyulangone.org

**Keywords:** discrimination, sexual orientation, e-cigarette, health disparities, polytobacco, sexual minorities, minority stress, tobacco, polytobacco

## Abstract

Research on whether sexual orientation discrimination is associated with multiple tobacco product use among sexual minority (SM) adults is limited. Thus, we explored the associations between sexual orientation discrimination and exclusive, dual, and polyuse among a subset of SM adults (18+) (*n* = 3453) using the 2012–2013 National Epidemiologic Survey on Alcohol and Related Conditions-III. We evaluated six indicators of prior-to-past-year sexual orientation discrimination separately and as a summary scale and defined past-year exclusive, dual, and polyuse based on cigarette, electronic nicotine delivery systems, other combustible (cigars and traditional pipe), and smokeless tobacco products. Using multinomial logistic regression, we estimated adjusted associations between sexual orientation discrimination and exclusive, dual, and polyuse. Experiencing discrimination in public places, being called names, and being bullied, assaulted, or threatened were associated with dual use, while experiencing discrimination when obtaining health care or insurance and when receiving health care were associated with polyuse. Each one-unit increase in the sexual orientation discrimination summary scale was associated with 5% and 10% higher odds of dual (95% CI: 1.01–1.10) and polyuse (95% CI: 1.02–1.18), respectively. To conclude, we advise health professionals to consider the salience of discrimination against SM adults and how these experiences lead to dual/polyuse.

## 1. Introduction

Tobacco use continues to burden the United States (US) as the nation’s leading behavioral cause of preventable morbidity and mortality [1,2]. While cigarette use has declined among US adults over the past several decades [3,4], use of other tobacco products such as electronic nicotine delivery systems (ENDS) have become more common [2,5]. As the tobacco industry continues to release more tobacco products into the market, researchers have become increasingly concerned about the use of multiple tobacco products, or dual use (use of two products) and polyuse (use of three or more products) [6,7]. Dual/polyuse is associated with higher frequency of use [8,9] and affects quit intentions and attempts [9,10,11], potentially leading to heightened susceptibility to nicotine dependence [12,13]. While understudied, dual/polyusers may be more at risk for physical health effects than exclusive users, given that use of non-cigarette tobacco products are also linked to respiratory and cardiovascular outcomes [14,15,16,17]. These factors make it important to study dual and polyuse patterns, including disparities in use, to determine the populations most at risk.

Recent evidence shows that sexual minority (SM) (i.e., individuals who identify as gay, lesbian, or bisexual and individuals with same-sex or bisexual attraction or relationships) adults use more tobacco products and experience greater health consequences of tobacco use compared to heterosexual adults [18,19,20,21,22]. For example, a meta-analysis reported that cigarette use is highest among bisexual (37.7%) and lesbian (31.7%) women, followed by gay (30.5%) and bisexual (30.1%) men, relative to heterosexual men (21.0%) and women (16.6%) [18]. Among a nationally representative sample, the odds of regular cigarette, e-cigarette, cigar, and hookah use were higher for adult SM women than adult heterosexual women [20]. The same was true for younger adult SM men for regular cigarette use, but not older adult SM men [20]. Many factors may be driving disparities in tobacco use by SM status, such as tobacco industry advertising and marketing, social norms surrounding tobacco use, and excess stress unique to sexual minority populations (i.e., minority stress) [23,24,25]. For example, the tobacco industry advertises and markets specific tobacco products toward SM populations to encourage and cement use behaviors [26,27,28]. Smoking has also been found to be more socially acceptable among SM adults, potentially driving disparities in tobacco use [29,30]. These factors likely interact with the minority stress that SM adults experience due to factors such as stigma, prejudice, and discrimination unique to SM populations [23,24].

The minority stress model is a conceptual tool that frames the experiences of SM populations and helps explain physical and mental health disparities among these groups [23,24]. Minority stress is greater than the general stress experienced by all persons, is omnipresent and manifests over the life course, and derives from social systems, institutions, and processes [23,24]. Minority stress can additionally impact the mental health of SM adults, with empirical studies suggesting relationships between stress and mental health outcomes [31]. Mental health problems related to stress are associated with substance use, including tobacco use, potentially through adverse stress-coping mechanisms [31,32,33,34]. In addition, social determinants unique to certain populations may help explain a distinct relationship between stress and behavior. A common, well-documented determinant that has widespread impact is discrimination, or inequitable treatment or harassment, that can lead to psychological stress responses associated with adverse health behaviors and outcomes [35]. A substantial proportion of SM adults may experience some form of discrimination in their lifetimes; one study demonstrated that more than 50% of SM adults experienced various forms of interpersonal discrimination such as microaggressions, sexual harassment, and violence [35]. In support of the minority stress model [23], studying sexual orientation discrimination may help elucidate the mechanisms driving disparities in tobacco use among SM populations.

Many studies have evaluated the relationship between racial/ethnic discrimination and tobacco use [36,37,38,39], but whether sexual orientation discrimination specifically leads to tobacco use is less well understood [40,41,42,43], especially in the context of dual/polyuse. Increased exposure to racial/ethnic discrimination is associated with tobacco use, particularly cigarette use [37,38,39]. A recent investigation has also established a connection between experiencing discrimination in general and other tobacco product use, such as e-cigarettes, cigars, pipes, and hookah [36]. Experiencing discrimination is additionally associated with the higher prevalence of tobacco use disorders, including among SM adults [40,41,42,43]. However, evidence on whether experiencing discrimination leads to dual/polyuse overall and among SM adults remains unexplored and is important to corroborate research suggesting that SM adults are at higher risk for behaviors related to nicotine dependence, tobacco use disorder, and other long-term health effects. Thus, this study aimed to examine the extent to which sexual orientation discrimination is associated with exclusive, dual, and polyuse among a nationally representative sample of SM adults in the US.

## 2. Materials and Methods

### 2.1. Data

We used data from the National Epidemiologic Survey on Alcohol and Related Conditions-III (NESARC-III), a nationally representative cross-sectional study on the adult (aged 18 and older) noninstitutionalized civilian population in the US. The data were collected between April 2012 and June 2013 and the study primarily aimed to evaluate the mental health status of US adults [44]. NESARC-III used the Alcohol Use Disorder and Associated Disabilities Interview Schedule-5 by the National Institute on Alcohol Abuse and Alcoholism, a semi-structured diagnostic interview to collect information on adults in their households. Information on household response and person response rates as well as the overall sample design are explained elsewhere [44]. Approval of the University of Michigan Institutional Review Board was obtained for this research.

### 2.2. Participants

The NESARC-III included 36,309 adults. SM status was captured via three dimensions: current same-sex or both-sex sexual attraction, lifetime same-sex or both-sex sexual behavior, and current sexual orientation (i.e., identified as gay, lesbian, or bisexual). Respondents who indicated that they belonged to a SM group, based on at least one of the dimensions, were asked questions related to sexual orientation discrimination (*n* = 4151). Respondents who had missing information on at least one sexual orientation discrimination measure (*n* = 691) or one tobacco product use measure (*n* = 7) were excluded in this study, resulting in an analytic sample size of 3,453. After these exclusions, additional data were not missing for the covariates of sex, race/ethnicity, urbanicity, and geographic region. For age, highest educational attainment, and annual household income, missing data were imputed by NESARC-III investigators [44]. Appendix A displays the proportions of SM status indicators for sexual attraction, sexual behavior, and sexual orientation overall and by sex among the full sample and analytic sample of SM adults.

### 2.3. Measures

#### 2.3.1. Sexual Orientation Discrimination

Derived from the Experiences of Discrimination Scale [45,46], NESARC-III included measures on sexual orientation discrimination assessing how often SM respondents experienced discrimination because they were assumed to be gay, lesbian, or bisexual in six scenarios at two time points (i.e., prior to a year ago, within the past year). We included six measures of prior-to-past-year sexual orientation discrimination in this analysis. Respondents were asked “(about) how often did you experience discrimination in” (1) “ability to obtain health care or health insurance coverage,” (2) “how you were treated when you got care”, (3) “public, like on the street, in stores or in restaurants”, (4) “ANY other situation, like obtaining a job or on the job, getting admitted to a school or training program, in the courts or by the police”, and “(about) how often were you” (5) “called names” and (6) “made fun of, picked on, pushed, shoved, hit, or threatened with harm”. Validity and reliability of the experiences of discrimination scale have been documented in previous studies [45].

For brevity, we shortened the labeling for sexual orientation discrimination in each situation, respectively, as: (1) obtaining health care or insurance, (2) receiving health care, (3) while in public places, (4) while in other situations, (5) called names, and (6) bullied, assaulted, or threatened. Response options for each scenario included: (0) never, (1) almost never, (2) sometimes, (3) fairly often, and (4) very often. We examined each scenario separately and as a summary scale (range 0–24), where the six were summed, similar to prior research using these measures [40,45,47]. The Cronbach’s alpha for the summary scale was 0.90, demonstrating excellent internal reliability and a similar value to other studies using these measures in NESARC-III [40,41].

#### 2.3.2. Exclusive, Dual, and Polytobacco Use

The following tobacco products were included in this analysis: cigarettes, ENDS (i.e., e-cigarettes, e-liquid), cigars, traditional pipe, and smokeless tobacco (SLT) (i.e., snuff, chewing tobacco). For each product, respondents who indicated that they used the product at least once in the past year were classified as past-year users. For this analysis, we derived four broader classifications of tobacco use among the five types of products present in NESARC-III: (1) cigarettes, (2) ENDS, (3) other combustibles (OC) (cigars and traditional pipe), and (4) SLT.

Similar to prior work [7], we created a 16-category variable of mutually exclusive tobacco use groups, or every possible combination of the four tobacco product use groups: (0) never/former (referent group), (1) exclusive cigarette, (2) exclusive ENDS, (3) exclusive OC, (4) exclusive SLT, (5) cigarette and ENDS, (6) cigarette and OC, (7) cigarette and SLT, (8) ENDS and OC, (9) ENDS and SLT, (10) OC and SLT, (11) cigarette, ENDS, and OC, (12) cigarette, ENDS, and SLT, (13) cigarette, OC, and SLT, (14) ENDS, OC, and SLT, (15) all four tobacco products. Using the 16-category variable, we derived an additional variable that captured never/former (category 0), exclusive use (categories 1–4), dual use (categories 5–10), and polyuse (categories 11–15). In this study, multiple tobacco product use refers to both dual and polyuse. We used the 16-category multiple tobacco product use variable for description (see Appendix A) and the four-category variable for regression models.

#### 2.3.3. Covariates

We included sociodemographic and geographic characteristics as covariates in the analysis. These included age, a quadratic term for age, sex (female, male), race/ethnicity (Hispanic, non-Hispanic White, non-Hispanic Black, another race/ethnicity), highest educational attainment (less than high school/high school graduate/GED vs. some college or more), annual household income (less than $25,000, $25,000 to $59,999, $60,000 or more), urbanicity (urban, rural), and US geographic region (Northeast, Midwest, South, West). Another race/ethnicity included American Indian/Alaska Native (AI/AN), Asian/Native Hawaiian/other Pacific Islander (A/NH/OPI), or multiracial adults, and these groups were aggregated due to small sample sizes. Urbanicity described urban or rural residency and was captured via county-level Rural-Urban Continuum Codes [44].

### 2.4. Statistical Analysis

We estimated the weighted prevalence of exclusive, dual, and polyuse overall and by sociodemographic and geographic characteristics. We compared weighted distributions and means across past-year tobacco use groups for each covariate using chi-square tests of independence or ANOVA tests. We calculated the means of each prior-to-past-year sexual orientation discrimination item and the sexual orientation discrimination summary scale overall and for each tobacco use group. The means were compared across tobacco use groups using ANOVA tests as well as pairwise t-tests comparing the following categories: (1) exclusive use vs. dual use, (2) exclusive use vs. polyuse, and (3) dual use vs. polyuse (see Appendix A).

We ran seven sets of unadjusted and covariate-adjusted multinomial logistic regression models, one for each prior-to-past-year sexual orientation discrimination scenario and the sexual orientation discrimination summary scale, to assess the association between each exposure and past-year exclusive, dual, and polyuse. Covariate-adjusted models controlled for mean-centered age, quadratic age, sex, race/ethnicity, highest educational attainment, annual household income, urbanicity, and geographic region. Age was centered by subtracting each value from its mean and to minimize collinearity between linear and quadratic age terms. The quadratic age term was added to the model due to the non-linear relationship between age and tobacco use.

#### 2.4.1. Sensitivity Analysis

We conducted a sensitivity analysis examining associations between past-year sexual orientation discrimination and past-30-day exclusive, dual, and polyuse to determine whether more recent discriminatory experiences were associated with multiple tobacco product use. For this analysis, we aggregated dual and polyuse due to limited sample sizes.

#### 2.4.2. Supplementary Analysis

We tested two-way interactions between each prior-to-past-year sexual orientation discrimination item and scale and sex based on prior work documenting sex differences in tobacco use among sexual minority populations [18,19,20]. We additionally explored how associations varied by SM status and sex using a subset of the analytic sample that included respondents who had complete information on the three SM dimensions (*n* = 3106). We created a SM-status-by-sex variable inspired by previous research [40] that contained four groups: (1) heterosexual-identified women with same-sex and/or both-sex sexual attraction and/or behavior (*n* = 1314), (2) heterosexual-identified men with same-sex and/or both-sex sexual attraction and/or behavior (*n* = 662), (3) SM-identified (i.e., lesbian or bisexual sexual orientation) women (*n* = 676), and (4) SM-identified (i.e., gay or bisexual sexual orientation) men (*n* = 454). We collapsed the gay/lesbian and bisexual sexual orientation categories to circumvent model convergence issues due to low sample size in the bisexual women and men groups. We fit an adjusted multinomial logistic regression model estimating the association between the prior-to-past-year sexual orientation discrimination scale and past-year exclusive and dual/polyuse with a two-way interaction term between discrimination and the SM-status-by-sex variable. Since the Wald *p*-value test for interaction was statistically significant (*p* = 0.038), we stratified the regression model to estimate stratum-specific relationships between the sexual orientation discrimination scale and past-year exclusive and dual/polyuse. Due to limited sample sizes, we collapsed dual/polyuse into one category for both supplementary analyses. We accounted for NESARC-III’s complex survey design and used Stata 16.1 for all analyses [48].

## 3. Results

### 3.1. Prevalence of Exclusive, Dual, and Polytobacco Use

Among SM adults, 26.7% were exclusive users, 7.3% were dual users, and 1.2% were polyusers (see Appendix A). The largest exclusive use group was exclusive cigarette use (24.3%); the largest dual use group was dual use of cigarettes and ENDS (4.8%); and the largest polyuse group was polyuse of cigarettes, ENDS, and OCs (0.5%).

### 3.2. Prevalence of Sociodemographic and Geographic Characteristics by Exclusive, Dual, and Polytobacco Use

The mean age of the analytic sample was 43.0 (SD: 0.5) (see Table 1). Overall, the sample had more female SM adults (60.6%) than male. Approximately two-thirds of the sample was non-Hispanic White (65.4%), while 14.0% was Hispanic, 12.9% was non-Hispanic Black, and 7.8% was another race/ethnicity. Nearly two-thirds of the sample had at least some college as their highest educational attainment (63.7%). Annual household income was nearly split across categories (<$25,000 at 34.5%, $25,000–$59,999 at 33.4%, $60,000 or more at 32.0%). Most SM adults lived in urban counties (83.9%), and in the South (33.7%) and West (26.8%) compared to the Northeast (19.9%) and Midwest (19.6%).

### 3.3. Mean Prior-to-Past-Year Sexual Orientation Discrimination by Exclusive, Dual, and Polytobacco Use

The overall mean of prior-to-past-year sexual orientation discrimination summary scale (range: 0–24) was 1.07 (95% confidence interval (CI): 0.94–1.21) (see Table 2). Means for each prior-to-past-year sexual orientation discrimination item (range: 0–4) were between 0.11 and 0.28, with the highest means reported for experiencing sexual orientation discrimination in public places (0.26) and being called names (0.28). Means for these two items also differed across tobacco use status, as did the means for the sexual orientation discrimination summary scale. Specifically, the mean was 0.99 (95% CI: 0.83–1.14) for never/former users, 1.09 (95% CI: 0.82–1.36) for exclusive users, 1.62 (95% CI: 1.08–2.16) for dual users, and 2.31 (95% CI: 0.83–3.78) for polyusers (*p* = 0.014).

### 3.4. Associations between Sexual Orientation Discrimination and Exclusive, Dual, and Polytobacco Use

In adjusted multivariable multinomial logistic regression models, we did not observe any statistically significant associations for exclusive use relative to never/former use (see Table 3). However, each one-unit increase on the 0–4 scale in sexual orientation discrimination while in public places (odds ratio (OR): 1.21, 95% CI: 1.02–1.43), for being called names (OR: 1.23, 95% CI: 1.04–1.45), and being bullied, assaulted, or threatened (OR: 1.29, 95% CI: 1.04–1.60) was associated with higher odds of dual use. In addition, each one-unit increase on the 0–24 scale in the sexual orientation discrimination summary scale was associated with 5% higher odds of dual use (95% CI: 1.01–1.10). For polyuse, each greater one-unit increase on the 0–4 scale in sexual orientation discrimination for obtaining health care or insurance (OR: 2.12, 95% CI: 1.31–3.42) and receiving health care (OR: 1.69, 95% CI: 1.05–2.71) resulted in elevated odds. Furthermore, each one-unit increase on the 0–24 scale in the sexual orientation discrimination summary scale was associated with 10% higher odds of polyuse (95% CI: 1.02–1.18).

### 3.5. Sensitivity Analysis

Our sensitivity analysis examining associations between past-year sexual orientation discrimination and past-30-day exclusive, dual, and polyuse (see Appendix A) revealed that each one-unit increase on the 0–24 scale in sexual orientation discrimination was associated with 7% higher odds of dual/polyuse, though the point estimate was only marginally statistically significant (95% CI: 1.00–1.15). In addition, every one-unit increase on the 0–4 scale in being bullied, assaulted, or threatened was associated with 30% higher odds of exclusive use (95% CI: 1.03–1.63) and 65% higher odds of dual/polyuse (95% CI: 1.18–2.31).

### 3.6. Supplementary Analysis

We did not observe statistically significant Wald test *p*-values for the interactions between prior-to-past-year sexual orientation discrimination measures and sex (see Appendix A). Thus, we did not disaggregate sex to further examine effect modification. Our subset analysis evaluating the association between prior-to-past-year sexual orientation discrimination and past-year patterns of tobacco product use by SM status and sex suggest that discrimination is significantly associated with dual/polyuse among heterosexual-identified men with same-sex and/or both-sex sexual attraction and/or behavior (OR: 1.23, 95% CI: 1.05–1.44).

## 4. Discussion

Our findings suggest that SM adults who report experiencing discrimination based on their sexual orientation status have elevated odds of dual and polyuse. Specifically, each one-unit increase on the 0–24 scale in sexual orientation discrimination was associated with 5% higher odds of past-year dual use and 10% higher odds of past-year polyuse. SM adults who experienced discrimination while in public places, by being called names, and by being bullied, assaulted, or threatened had higher odds of dual use, while SM adults who experienced discrimination while obtaining health care or insurance and receiving health care had higher odds of polyuse.

Results from our study empirically demonstrate the behavioral effects of minority stress in which SM adults experience stressors unique to their disadvantage and marginalization [23]. Even though the mean reported value for the sexual orientation discrimination scale was relatively low, suggesting that the overall sample of SM adults rarely experienced discrimination in the past year, we found that experiencing more discrimination was related to higher dual/polyuse for SM adults. The additional stress associated with discrimination may induce a coping response that leads to tobacco use and possibly the use of multiple tobacco products [23]. Prior work has demonstrated such a relationship among SM youth and adult populations, suggesting that tobacco prevention and cessation efforts focus on the salience of minority stress on use patterns [49,50].

We also observed noteworthy variation in associations between individual sexual orientation discrimination items and dual/polyuse. Being discriminated against while in public places, being called names, and being bullied, assaulted, or threatened were associated with dual use, while being discriminated against while obtaining health care or insurance or while receiving health care were associated with polyuse. These distinctions suggest that the mechanisms through which sexual orientation discrimination leads to the use of multiple tobacco products may differ by scenario. The stress response and associated negative affect that comes with being called names may differ from being denied or treated differently by health care professionals. For example, SM adults who experience maltreatment in healthcare settings may be more prone to using multiple tobacco products, especially if that treatment is related to tobacco cessation services among SM adults who already smoke. However, more context is needed, such as why SM adults are using multiple tobacco products, the extent to which reasons for use relate to stress-coping responses to prejudice/discrimination, differential/lack of access to tobacco use treatment services, other reasons such as social norms, or a combination of factors. These factors may help us further understand why being discriminated against while in community/work venues was associated with dual use, especially if we gained more insight on types of dual/polyuse (i.e., the specific product combinations). In our study, sample size limitations did not allow for further disaggregation of dual/polyuse, and we call for future work to examine these patterns in more detail.

Our findings corroborate other studies that have evaluated the extent to which sexual orientation discrimination leads to tobacco-related outcomes among SM adults. For example, McCabe et al. (2019) found that sexual orientation discrimination was associated with DSM-5 tobacco use disorders (TUD) among SM adults using the same dataset [40]. Our findings support these results, indicating that SM adults who experience more discrimination based on sexual orientation status may use multiple tobacco products, enhancing their risk for tobacco use disorders such as nicotine dependence [12,13]. In addition, McCabe et al. (2019) found that sexual orientation discrimination was associated with tobacco use across dimensions of SM status [40]. Results from our sensitivity analysis (see Appendix A) that focused on exclusive and dual/polyuse outcomes suggest that the association between sexual orientation discrimination and dual/polyuse was most pronounced among heterosexual-identified men with same-sex and/or both-sex sexual attraction and/or behavior. While these findings differ from previous research, they suggest that associations between discrimination and tobacco use outcomes among SM adult populations may vary by sexual orientation, attraction, and behavior. Further research that aims to disentangle associations between discrimination and dual/polyuse across dimensions of SM status is needed.

Another study by Evans-Polce et al. (2020) reported an association between past-year sexual orientation discrimination and TUD among adults aged 39.3–43.2 years [43]. However, older adults aged 48.7–50.0 years who reported sexual orientation discrimination had lower odds of TUD, highlighting how these associations may vary by age. Further work examining the impact of discrimination on tobacco use outcomes among sexual minority adults by sociodemographic factors such as age, sex, and race/ethnicity is necessary to better understand these relationships. Moreover, racial/ethnic discrimination is predictive of tobacco use disorders among SM adults [41]. While a plethora of research has linked racial/ethnic discrimination with substance use among racial/ethnic minority populations [37,38,39], few investigations have examined the intersection between discrimination attributed to sexual orientation and discrimination attributed to race/ethnicity. Future research may benefit from examining whether these collective forms of discrimination act together to induce a minority stress response among SM populations [51,52].

### Limitations

Our study has several limitations. First, data were collected between 2012 and 2013. Since then, the political and social environments surrounding SM experiences have changed, as well as the tobacco landscape of available and commonly used products. While tobacco use patterns have also changed in the past 10 years, our observed association between sexual orientation discrimination and dual/polyuse is still likely relevant today and can inform the current state of multiple tobacco product use, which is a growing public health concern. Second, all measures used in this study are self-reported and potentially subject to response biases such as recall and social desirability. Third, while our study examines the relationship between prior-to-past-year sexual orientation discrimination and past-year tobacco use, we cannot infer a causal relationship due to the cross-sectional nature of the data collection. There exists the possibility of reverse causation in which tobacco use leads to experiencing discrimination. For example, SM adults who are nicotine dependent may experience discrimination in public, at work, or while seeking healthcare due to their tobacco use status. However, we attempted to lag the exposure definition (prior-to-past-year) to the outcome (past-year) to mitigate the biases that arise from reverse causation. Relatedly, unmeasured confounding may have biased our effect estimates. For example, tobacco control policies (unmeasured in NESARC-III) may be associated with both sexual orientation discrimination and tobacco product use. Fourth, due to limited sample sizes, we did not have enough statistical power to estimate associations for past-30-day tobacco use, especially for polyuse separate from dual use. We also did not have enough statistical power to disaggregate SM adults by sexual attraction (same-sex or both-sex), behavior (same-sex or both-sex), and orientation (e.g., lesbian, gay, bisexual). Prior studies have demonstrated that tobacco use differs between lesbian/gay and bisexual sexual minorities [53,54], but we cannot observe these differences for dual/polyuse outcomes due to limited sample sizes.

## 5. Conclusions

This study examined the association between sexual orientation discrimination and exclusive, dual, and polyuse. Specific discriminatory experiences and a summary measure of sexual orientation discrimination were associated with dual and polyuse. These results provide support for the minority stress model in that excess stress from experiences related to stigma and discrimination may lead to substance use as a coping mechanism among SM adults. Overall, this study demonstrates the importance for health professionals and policymakers to consider how the discriminatory experiences may lead to adverse health behaviors and effects among SM adults. These considerations include calling for tobacco control efforts to employ consistent and systematic methods for promoting inclusivity in their programs or policies as well as culturally appropriate methods for prevention and treatment [25,55,56]. In addition, broader initiatives led by public health professionals, advocates, and policymakers focused on eliminating homophobic prejudice and discrimination across the US may reduce the burden of tobacco use, including multiple tobacco product use, among SM populations.

## Figures and Tables

**Table 1 ijerph-19-06305-t001:** Distributions of Participant Characteristics Overall and by Past-Year Exclusive, Dual, and Polytobacco Use (*n* = 3453).

	Past-Year Tobacco Product Use					*p* ^a^
Participant Characteristics	Overall (*n* = 3453)	Never/Former Use (*n* = 2217)	Exclusive Use (*n* = 965)	Dual Use (*n* = 233)	Polyuse (*n* = 38)	
Age, (mean ± SD)	43.0 ± 0.5	45.3 ± 0.5	40.3 ± 0.7	34.9 ± 1.0	32.1 ± 1.7	**<0.001**
Sex, *n* (%)						**0.028**
Female	2215 (60.6)	1427 (61.2)	626 (61.4)	144 (55.4)	18 (39.1)	
Male	1238 (39.4)	790 (38.8)	339 (38.6)	89 (44.6)	20 (60.9)	
Race/ethnicity, *n* (%)						**<0.001**
Hispanic	613 (14.0)	411 (14.7)	159 (12.5)	40 (14.6)	3 (4.6)	
Non-Hispanic White	1856 (65.4)	1161 (63.8)	525 (67.3)	142 (70.3)	28 (82.8)	
Non-Hispanic Black	760 (12.9)	471 (12.3)	245 (15.5)	39 (9.3)	5 (9.5)	
Non-Hispanic Other	224 (7.8)	174 (9.3)	36 (4.7)	12 (5.8)	2 (3.1)	
Highest educational attainment, *n* (%)						**<0.001**
<Highschool graduate/GED	1286 (36.3)	711 (30.4)	458 (48.1)	99 (43.5)	18 (53.0)	
Some college or more	2167 (63.7)	1506 (69.6)	507 (51.9)	134 (56.5)	20 (47.0)	
Annual household income, *n* (%)						**<0.001**
<$25,000	1441 (34.5)	811 (29.6)	502 (44.2)	106 (39.2)	22 (55.2)	
$25,000–59,999	1166 (33.4)	754 (32.4)	315 (35.3)	87 (38.3)	10 (18.7)	
$60,000 or more	846 (32.0)	652 (38.0)	148 (20.4)	40 (22.5)	6 (26.1)	
Urbanicity, *n* (%)						0.23
Urban	3041 (83.9)	1953 (83.8)	856 (84.6)	197 (80.0)	35 (94.3)	
Rural	412 (16.1)	264 (16.2)	109 (15.4)	36 (20.0)	3 (5.7)	
Geographic region, *n* (%)						0.09
Northeast	572 (19.9)	397 (21.3)	144 (18.7)	30 (14.7)	1 (1.6)	
Midwest	672 (19.6)	380 (18.1)	233 (22.4)	49 (21.8)	10 (26.5)	
South	1230 (33.7)	786 (33.0)	336 (34.6)	95 (37.0)	13 (36.0)	
West	979 (26.8)	654 (27.6)	252 (24.3)	59 (26.5)	14 (35.9)	

^a^ ANOVA or chi-square test comparing participant characteristics across tobacco product use groups; bold *p*-values are statistically significant (*p* < 0.05). SD: standard deviation.

**Table 2 ijerph-19-06305-t002:** Weighted Means of Prior-to-Past-Year Experiences of Sexual Orientation Discrimination Overall and by Past-Year Exclusive, Dual, and Polytobacco Use (*n* = 3453).

	Past-Year Tobacco Product Use					
Sexual Orientation Discrimination Items, Mean (95% CI) ^b^	Overall (*n* = 3453)	Never/Former Use (*n* = 2217)	Exclusive Use (*n* = 965)	Dual Use (*n* = 233)	Polyuse (*n* = 38)	*p* ^a^
Obtaining health care or insurance	0.11 (0.09, 0.12)	0.10 (0.08, 0.12)	0.09 (0.06, 0.13)	0.16 (0.06, 0.26)	0.47 (0.06, 0.89)	0.09
Receiving health care	0.11 (0.09, 0.13)	0.10 (0.08, 0.12)	0.10 (0.07, 0.14)	0.13 (0.05, 0.22)	0.36 (−0.06, 0.78)	0.18
While in public places	0.26 (0.22, 0.29)	0.23 (0.20, 0.27)	0.27 (0.20, 0.33)	0.38 (0.26, 0.50)	0.52 (0.06, 0.97)	**0.013**
While in other situations	0.14 (0.11, 0.16)	0.13 (0.10, 0.16)	0.14 (0.10, 0.18)	0.21 (0.10, 0.32)	0.20 (−0.03, 0.42)	0.16
Called names	0.28 (0.25, 0.32)	0.26 (0.22, 0.30)	0.30 (0.23, 0.38)	0.44 (0.32, 0.55)	0.46 (0.13, 0.79)	**0.003**
Bullied, assaulted, or threatened	0.18 (0.15, 0.21)	0.17 (0.13, 0.20)	0.18 (0.12, 0.24)	0.30 (0.19, 0.42)	0.30 (0.06, 0.55)	0.06
Sexual orientation discrimination scale ^b^	1.07 (0.94, 1.21)	0.99 (0.83, 1.14)	1.09 (0.82, 1.36)	1.62 (1.08, 2.16)	2.31 (0.83, 3.78)	**0.014**

^a^ ANOVA test comparing means of each discrimination item across tobacco product use groups. ^b^ Each sexual orientation discrimination item ranges from 0–4; the sexual orientation discrimination scale ranges from 0–24; bold *p*-values are statistically significant (*p* < 0.05).

**Table 3 ijerph-19-06305-t003:** Multivariable Multinomial Logistic Regression Models of Associations Between Prior-to-Past-Year Experiences of Sexual Orientation Discrimination and Past-Year Exclusive, Dual, and Polytobacco Use (*n* = 3453).

	Past-Year Tobacco Product Use ^a^					
	Exclusive Use		Dual Use		Polyuse	
Sexual Orientation Discrimination Items	AOR ^b^	95% CI	AOR ^b^	95% CI	AOR ^b^	95% CI
Obtaining health care or insurance	0.94	0.76, 1.16	1.23	0.90, 1.67	**2.12**	**1.31, 3.42**
Receiving health care	0.96	0.75, 1.22	1.10	0.78, 1.54	**1.69**	**1.05, 2.71**
While in public places	1.03	0.89, 1.19	**1.21**	**1.02, 1.44**	1.39	0.91, 2.12
While in other situations	1.03	0.84, 1.26	1.29	0.97, 1.71	1.24	0.74, 2.08
Called names	1.06	0.91, 1.23	**1.23**	**1.04, 1.45**	1.20	0.86, 1.67
Bullied, assaulted, or threatened	1.01	0.82, 1.26	**1.29**	**1.04, 1.60**	1.32	0.92, 1.90
Sexual orientation discrimination scale	1.00	0.96, 1.05	**1.05**	**1.01, 1.10**	**1.10**	**1.02, 1.18**

^a^ The outcome referent group: never/former use. ^b^ Adjusted odds ratios (AOR) and 95% confidence intervals (CI) adjusted for all participant characteristics. Bold AORs and 95% CIs are statistically significant (*p* < 0.05).

## Data Availability

This paper used limited access data obtained from the National Institute on Alcohol Abuse and Alcoholism (NIAAA). Data access must be requested at https://www.niaaa.nih.gov/research/nesarc-iii/nesarc-iii-data-access (accessed on 20 May 2022).

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
