# Peer review of "Sexual Orientation Discrimination and Exclusive, Dual, and Polytobacco Use among Sexual Minority Adults in the United States"

_ijerph, 2022, doi:10.3390/ijerph19106305_

Round 1

Reviewer 1 Report

This is a novel and important study of disparities in exclusive, dual, and polytobacco use among sexual minority populations. The authors use a rich data source to study the relationship between minority stress (discrimination in particular) and tobacco use, highlighting that discrimination related to sexual minority status is linked to dual- and polytobacco use.

The authors carefully and thoughtfully employ methodology that is both reflective of recent literature and responsive to their data as they untangle the relationship between discrimination and tobacco use. They do an excellent job of examining relationships among subgroups of the SM population among whom tobacco use is likely to vary and identifying relationships between intensity/place of discrimination and tobacco use. Their methods, results, and discussion are clear, convincing, and add important new knowledge to the literature on the role of minority stress among SM communities. I provide comments below to improve framing and clarity in the introduction.

Introduction:

Consider opening your introduction with disparities in tobacco use/outcomes among sexual minority communities and then moving on to the impact of dual- and polyuse.

Lines 38-39: Should this be “and LOWER quit intentions and attempts”?

Lines 53-54: Why are SM populations more receptive to directly interacting with tobacco-related messages than heterosexual populations? As someone familiar with SGM disparities but less familiar with tobacco use/tobacco use outcomes this isn’t quite clear to me. If the focus of the paper is the relationship between experiences of discrimination and tobacco use, I might shorten this section to mention the higher rates of advertising (which I think is very important) and then move quickly into minority stress and why it matters here.

Lines 60-61: I might consider starting this paragraph (or ending the previous paragraph) with an even broader summary sentence about minority stress theory (e.g., minority stress leads to/is linked to health disparities). Then you can name examples of minority stress (perhaps move from the end of the previous paragraph).

Lines 61-64: These definitions are useful but seem a bit out of place. It could help to contextualize with a bit more information about minority stress (as suggested above) or save these definitions for when you are actually describing how minority stress and discrimination are operationalized in the data.

Lines 74-76: These sentences make it sound like the link between discrimination (and arguably minority stress, though not sexual minority stress) and tobacco use have been studied. However, in the previous paragraph, the phrasing makes it seem like this direct link has not been studied (just that stress leads to worse mental health and worse mental health can lead to tobacco use). I think it makes sense to mention the relationship between minority stress and mental health when giving background on the minority stress model but it would be helpful to understand more clearly what is known (or not known) about the relationship between minority stress (or discrimination specifically) and tobacco use. If you want to comment on the role of mental health in the relationship between minority stress and tobacco use, then perhaps this piece needs to be expanded even further.

Methods:

Line 159: Did you have information on gender identity or sex besides male/female?

Author Response

This is a novel and important study of disparities in exclusive, dual, and polytobacco use among sexual minority populations. The authors use a rich data source to study the relationship between minority stress (discrimination in particular) and tobacco use, highlighting that discrimination related to sexual minority status is linked to dual- and polytobacco use.

The authors carefully and thoughtfully employ methodology that is both reflective of recent literature and responsive to their data as they untangle the relationship between discrimination and tobacco use. They do an excellent job of examining relationships among subgroups of the SM population among whom tobacco use is likely to vary and identifying relationships between intensity/place of discrimination and tobacco use. Their methods, results, and discussion are clear, convincing, and add important new knowledge to the literature on the role of minority stress among SM communities. I provide comments below to improve framing and clarity in the introduction.

            Response: We appreciate your kind words and feedback.

Introduction:

Consider opening your introduction with disparities in tobacco use/outcomes among sexual minority communities and then moving on to the impact of dual- and polyuse.

Response: We thank you for the suggestion. We decided to keep information on the overall tobacco use burden in the US in the first paragraph, and focus on disparities in tobacco use among sexual minority (SM) adults in the second paragraph. However, we have added a couple of sentences and a reference to the second paragraph about tobacco use disparities among SM adults. The new text reads, “For example, a meta-analysis reported that cigarette use is highest among bisexual (37.7%) and lesbian (31.7%) women, followed by gay (30.5%) and bisexual (30.1%) men, relative to heterosexual men (21.0%) and women (16.6%) [18]. Among a nationally representative sample, the odds of regular cigarette, e-cigarette, cigar, and hookah use were higher for adult SM women than adult heterosexual women [20]. The same was true for younger adult SM men for regular cigarette use, but not older adult SM men [20].”

Lines 38-39: Should this be “and LOWER quit intentions and attempts”?

Response: We have modified this part of the sentence to read “impacts quit intentions and attempts.” Generally, dual/poly use is associated with higher quit intentions/attempts than exclusive use but quit success rates are the same across exclusive and dual/poly use groups.

Lines 53-54: Why are SM populations more receptive to directly interacting with tobacco-related messages than heterosexual populations? As someone familiar with SGM disparities but less familiar with tobacco use/tobacco use outcomes this isn’t quite clear to me. If the focus of the paper is the relationship between experiences of discrimination and tobacco use, I might shorten this section to mention the higher rates of advertising (which I think is very important) and then move quickly into minority stress and why it matters here.

Response: Based on the reference we cited, SM populations may be more exposed to marketing/advertising related to tobacco and are actively searching for tobacco-related information, compared to heterosexual populations. Social acceptability of smoking differs between heterosexual and SM populations as well, which might explain differences in receptivity. However, we considered your advice and omitted this part of the text.

Lines 60-61: I might consider starting this paragraph (or ending the previous paragraph) with an even broader summary sentence about minority stress theory (e.g., minority stress leads to/is linked to health disparities). Then you can name examples of minority stress (perhaps move from the end of the previous paragraph).

Response: Thank you for the suggestion. We have included another sentence to the beginning of our third paragraph that reads, “The Minority Stress Model is a conceptual tool that frames the experiences of SM populations and helps explain physical and mental health disparities among these groups [23, 24].”

Lines 61-64: These definitions are useful but seem a bit out of place. It could help to contextualize with a bit more information about minority stress (as suggested above) or save these definitions for when you are actually describing how minority stress and discrimination are operationalized in the data.

Response: We described minority stress more per your previous suggestion and believe that the definitions that follow are now appropriately placed in the text.

Lines 74-76: These sentences make it sound like the link between discrimination (and arguably minority stress, though not sexual minority stress) and tobacco use have been studied. However, in the previous paragraph, the phrasing makes it seem like this direct link has not been studied (just that stress leads to worse mental health and worse mental health can lead to tobacco use). I think it makes sense to mention the relationship between minority stress and mental health when giving background on the minority stress model but it would be helpful to understand more clearly what is known (or not known) about the relationship between minority stress (or discrimination specifically) and tobacco use. If you want to comment on the role of mental health in the relationship between minority stress and tobacco use, then perhaps this piece needs to be expanded even further.

Response: We indicated that the relationship between racial/ethnic discrimination and tobacco use has been studied, but that the relationship between sexual orientation discrimination, or discrimination among SM populations, and tobacco use is understudied, especially in the context of dual/polyuse. After rereading our paragraph, we concluded that the mental health piece did not fit well where we originally had it and moved that part of the paragraph toward the beginning. We also added clarity to that sentence by including text about the stress-coping mechanism as a potential explanation for how excess stress leads to tobacco use and a sentence about social determinants of health. This new text reads, “Mental health problems related to stress are associated with substance use, including tobacco use, potentially through adverse stress-coping mechanisms [31-34]. In addition, social determinants unique to certain populations may help explain distinct relationship between stress and behavior.” We believe that the paragraph flows better now and sets up the following paragraph on what is known/not known about the relationship between sexual orientation discrimination and tobacco use and dual/polyuse.

Methods:

Line 159: Did you have information on gender identity or sex besides male/female?

Response: Unfortunately, we only had information on sex (male, female).

Reviewer 2 Report

Thanks for this very important topic precise analysis.

Introduction:

- “plague” might be to moral, another term could be used here

Limitation:

- is the use of tobacco is balanced by a better self care amongst SM population (better alimentation, more sport practices etc.) ?

That could be interesting to cross this study with others of this kind to get a more precise representation of global care practices amongst SM. Otherwise, great study, thanks.

Author Response

Thanks for this very important topic precise analysis.

Introduction:

“plague” might be to moral, another term could be used here

Response: We thank you for the suggestion and have changed “plague” to “burden.”

Limitation:

Is the use of tobacco is balanced by a better self care amongst SM population (better alimentation, more sport practices etc.) ?

Response: We are unsure whether there is a balance between the use of tobacco and other healthier lifestyle choices among SM populations, but whether other healthier lifestyle choices counteract the harmful effects of tobacco use is unlikely. Nevertheless, we believe these considerations are outside the scope of our present study.

That could be interesting to cross this study with others of this kind to get a more precise representation of global care practices amongst SM. Otherwise, great study, thanks.

Response: We agree that it would be interesting and important to investigate these relationships globally.

Reviewer 3 Report

  1. The data was obtained from 2012-to 2013, which was 10 years ago. Are there any potential changes during the past ten years? How likely are the results can still to reflect the current situation?
  2. In the method 2.2 participants, what is the difference between “current same-sex or both-sex sexual attraction “ and “current sexual orientation (i.e., identified as gay, lesbian, or bisexual)”? What is the logic for using three dimensions to capture SM status?
  3. I do not think “age” which is the essential demographic variable, should be imputed.
  4. What is the detailed procedure for shortening the scale? How can you make sure the short version would not affect the reliability and validity of the original scale?
  5. What does “e-liquid” mean?
  6. In the covariate-adjusted modules, why do you control both mean-centred age and quadratic age? What was the purpose of controlling mean-centred age and quadratic age, instead of simply controlling the actual age?
  7. If there is a non-linear relationship between age and tobacco use, why not considering to apply non-linear analysis, rather than amending age to quadratic age? Are there any previous criteria for amending the age variable (g., provide references)?

Author Response

The data was obtained from 2012-to 2013, which was 10 years ago. Are there any potential changes during the past ten years? How likely are the results can still to reflect the current situation?

Response: We realize that the data were collected nearly 10 years ago and have addressed these concerns in the Limitations section. We have updated that text slightly, which now reads, “Our study has several limitations. First, data were collected between 2012 and 2013. Since then, the political and social environments surrounding SM experiences have changed, as well as the tobacco landscape of available and commonly used products. While tobacco use patterns have also changed in the past 10 years, our observed association between sexual orientation discrimination and dual/polyuse is still likely relevant today and can inform the current state of multiple tobacco product use, which is a growing public health concern.”

Also, our use of data from 10 years ago highlights an issue with data collection. Few nationally representative surveys capture detailed information on sexual orientation discrimination, substance use, and sexual minority status. Our analysis emphasizes the need for improved data collection to understand the role of discrimination in contributing to tobacco use patterns in SM populations.

In the method 2.2 participants, what is the difference between “current same-sex or both-sex sexual attraction “ and “current sexual orientation (i.e., identified as gay, lesbian, or bisexual)”? What is the logic for using three dimensions to capture SM status?

Response: The three dimensions help us more exhaustively capture SM status among the sample. Please refer to our first supplementary table that shows weighted proportions of SM status dimensions by sex. This table shows that while most of the analytic sample identified as heterosexual, these adults were classified as a sexual minority due to having same-sex/both-sex sexual attraction and/or same-sex/both-sex sexual behavior. If we, for example, relied only on sexual orientation status to identify the SM population, we might underrepresent the experiences of the population.

The following paper also sufficiently outlines tobacco use in the context of the three dimensions of sexual orientation: McCabe SE, Hughes TL, Matthews AK, et al.: Sexual orientation discrimination and tobacco use disparities in the United States. Nicotine Tob Res 2019;21:523-531.

I do not think “age” which is the essential demographic variable, should be imputed.

Response: The imputation was undertaken by NESARC-III study investigators prior to our receipt of the data. As such, we were not able to alter the imputation process.

What is the detailed procedure for shortening the scale? How can you make sure the short version would not affect the reliability and validity of the original scale?

Response: We are not entirely sure of what you mean by shortening the scale. We analyzed each subscale and the summary scale in this study. We mentioned shortening the labels for each subscale and the summary scale for brevity (i.e., how we referred to the variables in the text), and perhaps the wording may have confused you into believing we shortened the measurement of the scales.

The following labels “1) ability to obtain health care or health insurance coverage, 2) how you were treated when you got care, 3) public, like on the street, in stores or in restaurants, 4) ANY other situation, like obtaining a job or on the job, getting admitted to a school or training program, in the courts or by the police, 5) called names, and 6) made fun of, picked on, pushed, shoved, hit, or threatened with harm” were shortened to “1) obtaining health care or insurance, 2) receiving health care, 3) while in public places, 4) while in other situations, 5) called names, and 6) bullied, assaulted, or threatened,” respectively.

What does “e-liquid” mean?

Response: E-liquid is the fluid that is put into e-cigarettes to create aerosols through a heating process. We included the term because it and “e-cigarettes” were used in the questionnaire to denote any electronic nicotine delivery system use.

In the covariate-adjusted modules, why do you control both mean-centred age and quadratic age? What was the purpose of controlling mean-centred age and quadratic age, instead of simply controlling the actual age?

Response: We explain in the Statistical Analysis section that we mean-centered age to reduce collinearity between this variable and quadratic age. Multicollinearity among two independent variables can result in unreliable effect estimates. Quadratic age was included in the models since there is a non-linear relationship between age and tobacco use.

If there is a non-linear relationship between age and tobacco use, why not considering to apply non-linear analysis, rather than amending age to quadratic age? Are there any previous criteria for amending the age variable (g., provide references)?

Response: We did not use non-linear models because age was not the primary exposure variable of interest, but rather a confounder.

Round 2

Reviewer 3 Report

The authors addressed my comments.